# YB-1 Is Altered in Pregnancy-Associated Disorders and Affects Trophoblast in Vitro Properties via Alternation of Multiple Molecular Traits

**DOI:** 10.3390/ijms22137226

**Published:** 2021-07-05

**Authors:** Violeta Stojanovska, Aneri Shah, Katja Woidacki, Florence Fischer, Mario Bauer, Jonathan A. Lindquist, Peter R. Mertens, Ana C. Zenclussen

**Affiliations:** 1Department of Environmental Immunology, Helmholtz-Centre for Environmental Research-UFZ-, 04318 Leipzig, Germany; florence.fisher@ufz.de (F.F.); mario.bauer@ufz.de (M.B.); 2Clinic of Nephrology and Hypertension, Diabetes and Endocrinology, Otto-von-Guericke University, 39120 Magdeburg, Germany; aneri.shah@ovgu.de (A.S.); jon.lindquist@med.ovgu.de (J.A.L.); peter.mertens@med.ovgu.de (P.R.M.); 3Medical Faculty, Otto-von-Guericke University, 39120 Magdeburg, Germany; katja.woidacki@med.ovgu.de; 4Perinatal Immunology, Saxonian Incubator for Clinical Translation, Medical Faculty, University of Leipzig, 04103 Leipzig, Germany

**Keywords:** cold shock protein, intrauterine growth restriction, preeclampsia, placentation, apoptosis, NF-κB

## Abstract

Cold shock Y-box binding protein-1 (YB-1) coordinates several molecular processes between the nucleus and the cytoplasm and plays a crucial role in cell function. Moreover, it is involved in cancer progression, invasion, and metastasis. As trophoblast cells share similar characteristics with cancer cells, we hypothesized that YB-1 might also be necessary for trophoblast functionality. In samples of patients with intrauterine growth restriction, YB-1 mRNA levels were decreased, while they were increased in preeclampsia and unchanged in spontaneous abortions when compared to normal pregnant controls. Studies with overexpression and downregulation of YB-1 were performed to assess the key trophoblast processes in two trophoblast cell lines HTR8/SVneo and JEG3. Overexpression of YB-1 or exposure of trophoblast cells to recombinant YB-1 caused enhanced proliferation, while knockdown of YB-1 lead to proliferative disadvantage in JEG3 or HTR8/SVneo cells. The invasion and migration properties were affected at different degrees among the trophoblast cell lines. Trophoblast expression of genes mediating migration, invasion, apoptosis, and inflammation was altered upon YB-1 downregulation. Moreover, IL-6 secretion was excessively increased in HTR8/SVneo. Ultimately, YB-1 directly binds to NF-κB enhancer mark in HTR8/SVneo cells. Our data show that YB-1 protein is important for trophoblast cell functioning and, when downregulated, leads to trophoblast disadvantage that at least in part is mediated by NF-κB.

## 1. Introduction

Adequate placenta development is a prerequisite of a successful pregnancy, while inadequate placentation is a feature of many pregnancy-associated disorders, including spontaneous abortion, intrauterine growth restriction (IUGR), and preeclampsia (PE) [1,2]. Although these disorders have complex pathophysiology with incompletely understood etiology, many share similar origins, such as inadequate trophoblast proliferation and shallow trophoblast invasion [3]. The placenta arises from the trophectoderm of the developing embryo, and, as soon as the embryo-derived trophoblasts adhere to the maternal endometrium, they start to proliferate, migrate towards the basal membrane, and invade into the surrounding endometrial stroma [4]. Here, they attain to remodel the spiral arteries under tightly balanced exposure to growth factors and cytokines derived from the surrounding immune and stromal cells [5].

Interestingly, the trophoblasts share several common characteristics with malignant cells, including proliferative, migratory and invasive features [6]. Moreover, the molecular traits of the trophoblasts, e.g., gene expression and cell response to extracellular stimuli, are similar to those found in malignant cells [7]. Hence, understanding of the molecular mechanisms underlying tumor growth and invasiveness might be of essential interest in understanding the trophoblast functionality, as well.

In several cancer types, YB-1 has been reported as a promoter of cell proliferation, migration, invasion, and inflammation and as an inhibitor of apoptosis [8,9,10,11]. YB-1 is encoded by the YBX1 gene, performs pleotropic functions, and contains a highly conserved cold shock domain. This domain has an extreme affinity to bind to DNA and RNA [12], which enables YB-1 to regulate the expression of numerous genes, such as the mechanistic target of rapamycin (mTOR), vascular endothelial growth factor (VEGF), signal transducer and activator of transcription 3 (STAT3), nuclear factor ’kappa-light-chain-enhancer’ of activated B-cells (NF-κB), and major histocompatibility complex class 2 genes (MHC2) and Notch homolog 3 (NOTCH3), which are involved in cell growth and metabolism, angiogenesis, inflammation, immune system evasion, and embryo development, respectively [13,14,15,16]. In addition to the functions in carcinogenesis, YB-1 has been suggested to play a role in embryo development, as well [17]. Throughout embryogenesis, it is highly expressed in the skeletal muscle, spleen and liver, and, after birth, its expression levels quickly decrease [17]. In YBX1-/- knockout embryos, the development advances usually up to embryonic day 10.5 (E10.5), and, afterwards, they exhibit severe growth retardation, neurological and pulmonary lesions, and are embryonically lethal by E18 [18,19], which indicates that YB-1 is necessary at late developmental stages. Recently, we characterized the effects of YB-1 deficiency on placenta development in vivo [20], where trophoblast-specific YB-1 deficient mice showed reduced implantation areas and negatively affected gross placental morphometry already at E10 [20]. This shows a direct involvement of YB-1 not only in placenta growth but also in implantation processes. Knowing that YB-1 is necessary for adequate placental development, we now aim to understand the participation of YB-1 in trophoblast functionality. In that order, firstly, we checked the relevance of YB-1 mRNA expression in different pregnancy-associated disorders using patient samples. Secondly, we performed targeted overexpression and downregulation analysis of human YB-1 in two different trophoblast cell lines. Ultimately, we investigated a possible underlying connection with trophoblast specific genes of proliferation, migration, invasion, and inflammation.

## 2. Results

### 2.1. YB-1 Expression Is Unaltered in Miscarriage Samples but Is Impaired in IUGR and PE Syndrome

To assess whether YB-1 is affected in early pregnancy events, we analyzed the YB-1 expression in placenta samples from spontaneous abortions and normally progressing pregnancies that were legally terminated (induced abortions), which served as controls. Overview of the baseline patients characteristics is given in Table 1. Unexpectedly, there were no changes in YBX1 expression levels between the groups (Figure 1A). Next, we checked the YBX1 levels in control term placentas and placentas obtained from late pregnancy events, such as preterm birth complicated by IUGR or PE syndrome. In IUGR placentas, YBX1 expression was significantly lower in comparison to controls (Figure 1B). This was, however, not the case in the group of patients with PE syndrome, where we detected increased YBX1 expression levels (Figure 1B). To further investigate whether YB-1 can be detected in serum from pregnant women in the last trimester, we quantified the YB-1 concentrations in a next set of patients. As expected, YB-1 serum concentrations were in line with the results obtained from the gene expression analysis in the placenta, meaning that IUGR pregnancies had lower YB-1 concentrations (Figure 1C). We observed a trend towards increased YB-1 serum concentrations in PE pregnancies, but the results did not reach statistical significance compared to the controls (Figure 1C). Our data suggest that YB-1 expression is not affected in the same way in different pregnancy-associated disorders with a possible role in later stages of pregnancy.

### 2.2. YB-1 Enhances Trophoblast Cell Proliferation

To address whether trophoblast growth is affected by YB-1, we overexpressed YBX1 in two different trophoblast cell lines. We used the choriocarcinoma JEG3 cell line and HTR8/SVneo, a first trimester immortalized trophoblast cell line. These two cell lines have different origins and are used to assess different trophoblast characteristics [21]. Overexpression of YB-1 (Figure 2A) results in proliferative advantage at 24 h for both cell lines (Figure 2B), without affecting the number of dead cells (Appendix A). Next, we assessed whether the effect of YB-1 on trophoblast proliferation is similarly mediated by adding recombinant YB-1 (rYB-1) protein to the culture. We tested several concentrations of rYB-1, and the effects were recorded after 24 h of exposure to the recombinant protein. At 5 μg/mL, rYB-1 induced proliferation in both trophoblast cell lines, while lower concentrations of rYB-1 showed proliferative capacity only for the HTR8/SVneo cell line (Figure 2C). Collectively, these data show that YB-1 positively affects trophoblast viability and proliferation.

### 2.3. Exposure to YB-1 Positively Affects Migration and Invasion of Trophoblasts

Next, we assessed the effect of YB-1 on trophoblast migratory abilities by performing wound healing assays with HTR8/SVneo cells. rYB-1 significantly increased the trophoblast migration (Figure 3A), and, similarly, the overexpression of YBX1 in HTR8/SVneo cells leads to increased migration and complete closure of the wound in 24 h (Figure 3B). Additionally, the invasion properties of HTR8/SVneo and JEG3 cells upon exposure to rYB-1 were assessed via transwell invasion assay. In line with the previous observations, the invasion rates of HTR8/SVneo cells were around 2-fold increased after 18-h exposure to rYB-1 (Figure 3C left panel), while almost no effect was seen regarding invasion properties of JEG3 cells upon stimulation with rYB-1 (Figure 3C, right panel). Hence, YB-1 stimulated migration and invasion in HTR8/SVneo trophoblast cells but not in JEG3 choriocarcinoma cells.

### 2.4. Loss of YB-1 Function Reduces Proliferation and Affects Apoptosis in Trophoblasts

Given that YBX1 overexpression positively affected the trophoblast phenotype, we next investigated whether YB-1 is crucial for overall trophoblast functionality. For this, we used a control and two shRNA YB-1 lentiviral vectors (shYB-1.1 and shYB-1.2), which were used to transduce HTR8/SVneo and JEG3 cell lines. Transduction efficacy and downregulation of YB-1 was confirmed by Western Blot. In both cell lines, the controls transduced with a scramble shRNA expressed endogenous YB-1, but, upon lentiviral transduction, with both shRNA YB-1 vectors, at least a 4-fold decrease in expression was observed (Figure 4A). As indicated in Figure 4B, downregulation of YB-1 provided a proliferative disadvantage of HTR8/SVneo as the population of live cells showed a 3-fold lower numbers at 72 h and more than a 7-fold lower numbers at 96 h post-seeding of the cells (Figure 4B, upper panel). A similar trend was observed when JEG3 cells were used, where the number of live cells decreased significantly at 72 h post-seeding (Figure 4B, lower panel). Additionally, as apoptosis is an important regulator of growth in trophoblast cells, we detected the rate of apoptosis in intact HTR8/SVneo cells using a caspase 3/7 fluorescent assay. Interestingly, YB-1 downregulation resulted in a 3-fold and 6-fold increase of caspase 3/7 activated cells (Figure 4C). Furthermore, we tested the expression of genes related to apoptosis in trophoblast cells (Figure 4D). Levels of mRNA of B-cell lymphoma 2 (BCL2), caspase 3 (CASP3), and FOS Like 1 (FOSL1) were upregulated in HTR8/SVneo YB-1 downregulated cells (Figure 4D, left panel). No changes were observed for BCL2 Associated X (BAX), caspase 9 (CASP9), and galectin-3 (LGALS3) (Figure 4D, left panel). In comparison, in JEG3 YB-1 downregulated cells only, BCL2 was significantly upregulated related to controls (Figure 4D, right panel). These data indicate that, in vitro, YB-1 downregulation results in a trophoblast growth disadvantage that is further associated with alterations in apoptosis-relevant genes.

### 2.5. Loss of YB-1 Affects Trophoblast Functionality by Modulation of Genes Involved in Cell Migration and Invasion

Decreased proliferation rates due to YB-1 downregulation may lead to decreased migration/invasion of trophoblast cells. In order to confirm that the impaired migratory and invasive capability of trophoblasts was not a side effect of decreased proliferation, we quantified the cell viability of YB-1 downregulated cells after 24 h in culture with an MTT assay. Both cell lines showed more than 80% cell viability compared to the controls treated with scramble shRNA (Figure 5A). Next, we evaluated the migratory properties of YB-1 downregulated HTR8/SVneo cells, and, as depicted in Figure 5B, it resulted in decreased migration in comparison to control HTR8/SVneo cells. Moreover, the invasion properties of YB-1 downregulated HTR8/SVneo and JEG3 cells were also decreased to different degrees (Figure 5C). Additionally, we analyzed the expression levels of genes involved in the promotion and inhibition of cell migration and invasion (Figure 5D). While matrix metalloproteinase 9 (MMP9) was differentially expressed in both HTR8/SVneo shYB-1 groups, the inhibitors of matrix metalloproteinases, tissue inhibitor of metalloproteinase 1 (TIMP1) and serpin family E member 1 (SERPINE1) were both significantly upregulated (Figure 5D, left panel). Additionally, there were no changes in the expression levels of macrophage migration inhibitory factor (MIF) (Figure 5D, left panel). In JEG3 cells, matrix metalloproteinase 2 (MMP2) showed disproportion in expression after YB-1 downregulation with the two shRNAs against YB-1, while MMP9, TIMP1, and MIF levels were similar to the controls (Figure 5D, right panel). Nevertheless, mRNA levels of SERPINE1 were significantly upregulated in JEG3 cells with downregulated YB-1 expression (Figure 5D, right panel). Moreover, in HTR8/SVneo YB-1 downregulated cells, NOTCH1 and NF-κB were both significantly upregulated, and signal transducer and activator of transcription 3 (STAT3) showed no changes between the groups, while NOTCH3 was downregulated (Figure 5D, left panel). As for JEG3 YB-1 downregulated cells, only NOTCH1 and STAT3 were significantly upregulated, and no changes in the expression of NOTCH3 were observed, while NF-κB showed a trend towards decreased expression (Figure 5D, right panel). Taken together, these results indicate that YB-1 downregulation promotes trophoblast dysfunctionality with effective regulation of downstream targets.

### 2.6. YB-1 Mediates IL-6 Secretion and Directly Binds to NF-κB Regulatory DNA Regions

YB-1 is implicated in inflammation and cytokine production. In the previous experiments, we showed that YB-1 downregulation altered the expression of NF-κB, NOTCH1, and NOTCH3, which are important modulators of the interleukin (IL-6) signaling pathways [22,23]. To investigate whether YB-1 downregulation also translates into changes in cytokine secretion, cell supernatants from YB-1 downregulated HTR8/SVneo and JEG3 cells were analyzed for interleukin 6 (IL-6) secretion. Interestingly, while YB-1 downregulated JEG3 cells showed similar IL-6 concentrations as the controls, HTR8/SVneo cells with downregulated YB-1 expression showed around a 40-fold increase in secreted IL-6 concentrations (Figure 6A).

Since YB-1 has translational, transcriptional, and chromatin binding abilities, we sought to investigate whether the genes that are significantly changed after YB-1 downregulation and are implicated in IL-6 secretion are functional targets of YB-1 under non-perturbed conditions. We tested the binding areas by designing CHIP-qPCR primers (Table 2) at the loci that are enriched for H3K27Ac. We chose these regions as potential binding sites for YB-1 as they indicate activation of transcription and are potential enhancer markers [24]. As a control, we confirmed that YB-1 antibody does bind to the YB-1 H3K27ac enriched area, but it does not bind to the selection position 2, which is an H3K27ac poor area (Figure 6B, lower panel). For NOTCH3, YB-1 does not bind throughout the selected H3K27ac enriched loci, which we identified as possible target marks (Figure 6C). Interestingly, we observed that YB-1 has a rudimentary binding to the chosen position 1, which lies at the exon 1 region of NF-κB in HTR/SVneo cell line (Figure 6D). Taken together, these data suggest that YB-1 might collaborate with NF-κB to induce impaired trophoblast phenotype.

## 3. Discussion

YB-1 plays an essential role in tumorigenesis by regulating cell proliferation, inflammation, migration, invasion, and apoptosis via relevant pathways [25]. During pregnancy, these processes are also crucial for the normal development of the placenta and changes in YB-1 expression might be involved in impaired placenta growth and/or functionality. We have recently shown that heterozygous YBX1 mice and mice with trophoblast-specific YBX1 deficiency display placental abnormalities with subsequent fetal growth retardation [20]. The present study demonstrates that pregnancy-associated disorders are associated with altered YB-1 concentrations in tissue and blood samples. Our experiments comprising overexpression and downregulation of YB-1 in trophoblasts show that YB-1 is an important regulator of their functionality.

Discordant YB-1 in pregnancy might lead to defective placentation and subsequent development of pregnancy-associated disorders. To date, we are the first to report results on YBX1 expression changes in patients suffering from spontaneous abortion, preterm birth, or PE syndrome. We did not observe any differences in placental YBX1 expression between subjects having a spontaneous abortion or induced termination of pregnancy. Thus, it suggests that YB-1 may not be particularly relevant in early events of pregnancy. However, in pregnancies complicated by preterm birth and intrauterine growth restriction, YB-1 transcript levels and serum concentrations were significantly decreased compared to controls. Thus, it indicates that YB-1 may have a considerable role in trophoblast physiology in the later stages of pregnancy. Surprisingly, in PE patients, YBX1 transcript levels were increased, but we could not confirm this in the few obtained matching serum samples. This discrepancy may be attributed to the slightly different pathophysiological traits underlying these disorders [26]. Preeclampsia, among defective placentation, is also characterized with angiogenic imbalance due to increased concentrations of antiangiogenic factors, such as soluble fms-like tyrosine kinase-1 (sFlt-1) and soluble endoglin (sEnd), and decreased concentrations of vascular endothelial growth factor (VEGF) and placental growth factor (PlGF) [27,28]. We speculate that this imbalance might, in turn, lead to a compensatory mechanism, such as an increase in YB-1 levels, which promotes angiogenesis [29]. Although it is known that YB-1 regulates the levels of angiogenic factors, it is yet to be elucidated whether angiogenic imbalance can trigger YB-1 upregulation. This might be the focus of subsequent studies, but it is not the main aim of the present one and will not be further discussed.

To investigate the impact of elevated YB-1 levels on the trophoblast phenotype, we used two different approaches: YBX1 overexpression and direct exposure to rYB-1 protein in the two most widely used trophoblast cell lines. The first one aims at understanding the significance of YBX1 expression in the trophoblast itself. The second one rather addresses the contribution of secreted YB-1 by other cells present at the feto-maternal interface [30]. Transient YBX1 overexpression led to an increased proliferation of both cell lines, and promotion of migration in HTR8/SVneo cells. Direct exposure to YB-1 resulted in similar outcomes, confirming the positive effect of this cold shock protein in stimulating the proliferation and migration of trophoblasts. Additionally, we tested the invasive properties of both cell lines in the presence of rYB-1, and, while no changes were registered for JEG3 cells, HTR8/SVneo cells increased their invasive activity upon YB-1 addition to the culture. The upregulation of trophoblast proliferation, migration, and invasion linked to YB-1 is in accordance with several previous studies using other cell types which demonstrated that overexpression of YBX1 acts as a potent enhancer of cellular functions [29,31]. In regard to the discrepancy we see in the affinity of YB-1 to the different trophoblast cell lines, we link this to the origin of the cell lines. The HTR8/SVneo cell line is a first trimester extravillous trophoblast cell line that was immortalized with an origin-defective simian virus 40 construct and was revealed that consists of two distinct populations; epithelial and mesenchymal [32]. Given that YB-1 is known to physically and functionally interact with the viral regulatory protein T- antigen [33], we speculate that this is the reason for the more pronounced effect in the HTR8/SVneo cell line compared to JEG3 cell line that is derived from cancerous tissue.

Normal placental development dependent not only on controlled cell proliferation and differentiation but also on programmed cell death that occurs as a rate-limiting proliferation factor [34]. Likewise, in normal pregnancies, trophoblast apoptosis increases as the pregnancy progress and in pregnancy-associated disorders, such as PE and IUGR, a greater incidence of apoptosis is observed [35]. The central players of apoptosis are the caspases, however many other genes are involved in this process [36]. Our study shows that YB-1 downregulated cells possess increased caspase activity and upregulated expression of the pro-apoptotic molecules BCL2 and FOSL1 in HTR8/SVneo cells. This clearly demonstrates that the absence of YB-1 results in excessive apoptosis in HTR8/SVneo cells via upregulation of several genes that propagate the death signal.

One essential feature for optimal placenta development is the ability of the trophoblasts first to migrate and then invade into the surrounding tissue. To date, YB-1 has been fundamentally implicated in the migration and invasion processes in several types of cancers, including breast cancer [37], lung cancer [38], melanoma [39], and spinal chordoma [40]. YB-1 downregulation leads to reduced invasive and migratory abilities of tumorigenic cells, consistent with changes in mRNA levels of genes necessary for cell proliferation and invasion, such as zinc finger protein SNAI1, NF-κB, MMP2, etc. [41]. This is also in line with our results, where we show that stable downregulation of YB-1 in trophoblast cell lines leads to a defective trophoblast phenotype that is impaired in proliferation, migration, and invasion. Furthermore, the invasive trophoblast ability is dependent on balanced protease activation and inhibition. While trophoblast invasion is mediated via MMPs, and namely MMP2 and MMP9, the inhibition is dependent on TIMP1 and SERPINE1 [42]. In our study, we show that the tissue inhibitors of matrix metalloproteinases TIMP1 and SERPINE1 were upregulated, and this may contribute to the decreased invasive profile of trophoblast cells.

Trophoblasts are known to secrete cytokines, and IL-6 is amongst the most abundantly produced factors [43]. IL-6 acts as a pro- and anti-inflammatory cytokine and modulates immune responses, angiogenesis, and trophoblast proliferation, migration, and apoptosis [44]. Moreover, cytotrophoblast cells express high IL-6 mRNA levels, which are further stimulated by IL-6 itself [45]. However, excessive production of IL-6 has been also implicated in the development of several pregnancy-associated disorders [46,47]. In the present study, we show that indeed, HTR8/SVneo cells with downregulated YB-1 secrete excessively high concentrations of IL-6. While the cytokine production is a tightly controlled process, the signaling pathways regulating IL-6 secretion are still not completely disclosed. Independently, it was reported that NOTCH3 and NF-κB could mediate the induction of IL-6 production in malignant cells [48] and in macrophages [22]. When determining the expression of NF-κB and NOTCH3 in our YB-1 downregulated HTR8/SVneo, we came to the conclusion that the expression levels were differentially affected. This provides evidence that YB-1 interacts with inflammation relevant genes that, in turn, can also perturb the trophoblast functionality.

Several studies have shown that YB-1 is required for NF-κB activation either via IL-1βR [49] or via TNFR1 [50]. Here, we show that, in unaltered conditions, YB-1 has the affinity to directly bind to a region that encloses the first exon of the NF-κB gene, which is also an H3K27ac enriched region as depicted in the UCSC genome browser [51]. This suggests that NF-κB can act as a functional target of YB-1. Knowing that NF-κB is involved in the regulation of several cellular processes, including proliferation, differentiation, apoptosis, oxidative stress, and inflammation [52], it is of no surprise that this protein is critically implicated in the placentation, as well [53]. Previously, it was shown that NF-κB can modulate the placenta development either via regulation of factors important for trophoblast invasion, such as MMP2, MMP9, and SERPINE1 [54], or via regulation of the feto-maternal vascularization through cytokines (IL-6, IL-8) [55] and angiogenic proteins (PlGF, VEGF, sEnd) [54,56]. Hence, the detrimental roles that we observe for YB-1 in trophoblast function may be, in part, mediated by NF-κB.

## 4. Conclusions

Our findings indicate that YB-1 acts as a potent regulator of trophoblast functionality via changes in the molecular footprint of genes involved in proliferation, apoptosis, migration, invasion, and inflammation. Moreover, we showed that YB-1 could directly bind to the NF-κB gene, which, in turn, can shed light on the YB-1 involvement in pregnancy-associated disorders.

## 5. Materials and Methods

### 5.1. Sampling and Ethical Approval

Placental tissue was collected from induced (elective) pregnancy terminations and spontaneous abortions or shortly after birth and was processed for RNA isolation. Peripheral blood samples were obtained from healthy pregnant or pregnant patients admitted to hospital due to preterm birth or development of PE. Inclusion criteria for PE were: singleton pregnancy, hypertension (diastolic blood pressure ≥ 90 mmHg on at least two occasions), and proteinuria (urine dipstick > 1+ (≥30 mg/dL) on at least two occasions, and protein: creatinine ratio of ≥0.35 or 24-h urine protein concentration ≥ 300 mg). For IUGR patients, the inclusion criteria were the following: singleton fetus, normotensive (120–80 mmHg), estimated fetal weight, and/or abdominal circumference <10th percentile. Exclusion criteria for all patients recruited in this study were: multiple pregnancy, presence of congenital infections and chromosomal defects, and autoimmune disorder of the mother. The clinical study was approved by the ethics board at the University of Magdeburg with reference number EK28/08. All subjects provided written informed consent, and the study was performed in accordance with the Declaration of Helsinki. Subject characteristics are presented in Table 1.

### 5.2. Cell Lines

The human choriocarcinoma cell line JEG3 (DMZO, Braunschweig, Germany) was cultured in Dulbecco’s modified Eagle’s medium (DMEM) (Invitrogen, Karlsruhe, Germany) supplemented with 10% fetal bovine serum (FBS, Biochrom, Berlin, Germany) and 100 nmol/L penicillin/streptomycin (Invitrogen, Karlsruhe, Germany). The immortalized human extravillous cytotrophoblast cell line HTR8/SVneo (ATCC, CRL-327) was cultured in Roswell Park Memorial Institute (RPMI) 1640 medium, supplemented with 10% fetal bovine serum (FBS, Biochrom, Berlin, Germany), 100 nmol/L penicillin/streptomycin (Invitrogen, Karlsruhe, Germany), 100 nmol/L MEM nonessential amino acids (Invitrogen, Karlsruhe, Germany), 1 mmol/L sodium pyruvate (Invitrogen, Karlsruhe, Germany), and 10 mmol/L HEPES (Invitrogen, Karlsruhe, Germany). Both cell lines were cultured at 37 °C with 5% CO${}_{2}$ and humidified atmosphere and subcultured using 0.05% Trypsin-EDTA (Invitrogen, Karlsruhe, Germany). The cell lines were regularly tested for absence of mycoplasma infection.

### 5.3. YB-1 Detection in Serum

Whole blood was collected in SST II Advance tubes (BD Vacutainer) and was allowed to clot for 30 min at room temperature. Afterwards, the samples were centrifuged at 2000 rpm for 10 min at 4 °C. Serum was collected, aliquoted, and stored at −80 °C until further analysis. Only samples that were not hemolyzed were used for the analysis. YB-1 serum concentrations were detected using ELISA method (Genway, Wuhan, China) following manufacturer’s instructions.

### 5.4. Recombinant YB-1 Protein Harvest and Purification

Seventy percent confluent HEK293T cells were transduced with pcDNA/Flag YB-1 sequence using Ca-phosphate-DNA precipitates. After 48 h of transduction, the cells were harvested in RIPA-lysis buffer (50 nM Tris, 150 nM NaCl, 1% Nonidet P40, 0.25% sodium deoxycholate, 1 mM EDTA, 1 mM Na${}_{3}$VO${}_{4}$, 1 mM NaF), supplemented with protease inhibitors (Complete EDTA-free cocktail tablet, Roche, Mannheim, Germany), and were immunoprecipitated using anti-DYKDDDDK G1 affinity resin (Genscript, Piscataway, NJ, USA) and binding buffer (50 mM Tris-HCl pH 7.4; 1 mM EDTA, 150 mM NaCl, 1% Triton X-100). The next day, the samples were eluted with 100 μg/mL FLAG-peptide stock (Sigma-Aldrich, St. Louis, MO, USA) and dialyzed with 40% Polyethylene glycol 20,000 (Roth, Karlsruhe, Germany). Protein concentration was determined using BioRad DC protein assay (BioRad, Hercules, CA, USA).

### 5.5. YBX1 Overexpression

JEG3 and HTR8/SVneo cells were transfected using Lipofectamine 2000 (Life Technologies, Carlsbad, CA, USA), according to the manufacturer’s protocol. In short, 70% confluent cells were transiently transfected with 1 μg control FuGW-eGFP or 1 μg FuGW-eGFP-YBX1 plasmid DNA. FUGW was a kind gift from David Baltimore (Addgene plasmid 14883; http://n2t.net/addgene:14883; RRID:Addgene 14883). After 24 h, the transfection efficacy was inspected by detection of GFP signal by fluorescence microscopy.

### 5.6. Lentiviral Transduction of YB-1

Downregulation of YB-1 was performed as previously described [50]. In short, the control plasmid pLKO and two different pLKO-YB-1.1 and pLKO-YB-1.2 shRNA were obtained from Sigma-Aldrich (shRNA: CCGGCCAGTTCAAGGCAGAAATATCTCGAGATA TTTACTGCCTTGAACTGG-TTTTTG). Two micrograms of YB-1 construct, 1 μg psPAX2, and 1 μg pVSV-G with calcium phosphate precipitates was used for lentiviral transduction of human embryonic kidney HEK 293T cells. Virus containing supernatants were then added to the target cells JEG3 and HTR8/SVneo. Stably transduced cell lines were selected using puromycin (1.5 μg/mL) for 7 days. Cells were harvested 3 and 7 days after transduction to confirm the YB-1 downregulation by Western Blot analysis.

### 5.7. Protein Isolation and Western Blot Analysis

Cells were lysed in RIPA buffer (50 mM Tris-HCl, 150 mM nonidet P-40, 1 mM sodium deoxycholate, 1 mM EDTA, and 1 mM Na${}_{3}$VO${}_{4}$) containing protease inhibitor cocktail (Complete EDTA-free cocktail tablet, Roche, Mannheim Germany) at 4°C for 30 min. Protein concentrations were quantified using Bio-Rad protein assay (BioRad, Hercules, CA, USA). Only the samples that had good protein yield were used for the study. Proteins were detected using primary antibodies anti-YB-1 (Eurogentec, Liège, Belgium) and anti-vinculin (Santa Cruz, Dallas, Texas, USA) was used as a loading control. Secondary antibodies coupled to horseradish peroxidase (Southern Biotech, Birmingham, AL, USA) were used for immuno-detection. The detection was performed using Pierce ECL Western blotting substrate (Thermo Fischer Scientific, Waltham, MA, USA).

### 5.8. Functional Assays

Proliferation was assessed by the trypan blue exclusion assay using Neubauer chambers and manual quantification of cells. For the cell migration assay, 20,000 cells were plated in 24-well plates, and, after 24 h, scratch was performed with 200 μL pipette tip. Pictures were taken after 24 h, and the distances crossed were measured using an electronic grid. The mean value for the controls was set to 100%, and the data are expressed as percentages of the control value. Transwell invasion assay was conducted in 24-well plate with 8 μm pore size Transwell inserts (Corning, Durham, NC, USA). Insert membranes were precoated with 50 μl of growth factors reduced Matrigel (Corning, Bedford, MA, USA) at a concentration of 0.5 μg/mL for 1 h at 37 °C. JEG3 or HTR8/SVneo (25,000 cells/100 μL) were resuspended in serum free medium with 1 or 5 μg/mL rYB-1 added to the upper part of the Transwell chamber. In the lower part of the chamber, complete medium was added with 10% FBS as chemoattractant. Cells were incubated for 18 h and the Transwell membranes were stained with 0.2% Crystal Violet (Sigma-Aldrich, Germany). Quantification of cells on the underside of the filter were counted with brightfield microscope for average of 10 picture fields at 20× magnification total.

### 5.9. RNA Isolation and qPCR Analysis

Total RNA was isolated using TRIzol (Life Technologies, Carlsbad, CA, USA), following manufacturer’s protocol. Quantity and purity of RNA were determined using Infinite F200 Nanoquant (Tecan, Grödig, Austria). The cDNA synthesis was carried out with 800 ng of total RNA by using Im Prom-II™ Reverse Transcription System (Promega, Mannheim, Germany). Real-time quantitative PCR was carried out on a LightCycler 480 System (Roche Applied Science, Mannheim, Germany) with the following cycling program: 2 min at 50 °C, 10 min at 95 °C, followed by 35 cycles of 95 °C for 15 s, 1 min at 60 °C, and 70 °C for 5 s. All reactions were performed in duplicates. Primers and UPL probes were designed and selected by the Universal Probe Library Assay Design Center (http://qpcr.probefinder.com/organism.jsp). Ct values and the expression of the reference gene GAPDH was uniform among the groups. The data was analyzed using the ddCT method.

### 5.10. IL-6 Quantification

Cell supernatants from trophoblast cell lines with downregulated YB-1 expression were collected, spun at 2000 rpm for 5 min at 4 °C, and stored at −80 °C until further analysis. IL-6 quantification was obtained with human IL-6 ELISA detection assay (R&D Systems, Minneapolis, MN, USA) according to manufacturer’s instruction. Absorbance was measured using Infinite F200 microplate reader (Tecan, Grödig, Austria).

### 5.11. Caspase Activity Assay

HTR8/SVneo control and YB-1 downregulated cells were plated in 24-well plate at a density of 20,000 cells per well and incubated for 48 h under standard cell culture conditions. The caspase activity was studied using CellEvent™ Caspase-3/7detection reagent (Invitrogen, Eugene, OR, USA). After 30 min of incubation, fluorescence was observed using fluorescent microscope (KEYENCE BZ-X800, Osaka, Japan), coupled with a confocal module. Excitation and absorption wavelength were 360/40 nm and 470/40 nm, respectively. Nuclei were stained with Hoechst 33342 (Invitrogen, Eugene, OR, USA). The intensity of fluorescence was analyzed with respective KEYENCE BZ-X800 analysis software.

### 5.12. MTT Assay

One thousand cells were plated onto 96-well plates, and 3-(4,5-dimethylthiazol-2-yl)-2,5-dipheny ltetrazolium bromide (Sigma-Aldrich, St. Loius, MO USA) was added after 24 h at a final concentration of 5 mg/mL. After incubation at 37 °C and 5% CO${}_{2}$ for 3 h, the MTT was removed and MTT formazan crystals were dissolved in 150 μL of DMSO. Absorbance at 590 nm was determined on an automatized microtiter plate reader (BioTek Synergy HT, Watertown, MA, USA).

### 5.13. Chromatin Immunoprecipitation Assay (CHIP)

Chromatin was isolated from 1x10${}^{7}$ JEG3 cells. In short, cells were cross-linked with 1% formaldehyde for 10 min and resuspended in SDS buffer (5 M NaCl, 1 M Tris-HCl, pH 8,1; 0.5 M EDTA pH 8,0; 10 M NaN${}_{3}$, 10% SDS) with added protease inhibitor (Complete EDTA-free cocktail tablet, Roche, Mannheim, Germany), and stored at −80 °C until further analysis. After defrost and centrifugation, samples were resuspended in IP buffer (66.7 mM Tris–HCl, 100 mM NaCl, 5 mM EDTA, 0.2% NaN${}_{3}$, 1.67% Triton-X-100, and 0.33% SDS) and sonicated 5 times for 30 s. Chromatin shearing was checked by reverse cross link reaction of 2 h on 65 °C at 850 rpm using reverse cross link buffer (Tris EDTA buffer 1x, SDS 20%, 5 M NaCl). One microgram of anti-YB-1 recombinant antibody (EP2708Y, Abcam, Cambridge, UK), or IgG from rabbit serum (I8140, Sigma-Aldrich, St. Louis, MO, USA), was used per immunoprecipitation. nProtein A Sepharose beads (GE Healthcare, Danderyd, Sweden) were used to pull-down the immune complexes. Wash of the beads and samples complexes was performed with High Salt Buffer 500 mM (1% Triton X-100, 0,1% SDS, 150 mM NaCl, 2 mM EDTA pH 8.0, 20 mM Tris-HCl) and 1x Tris-EDTA buffer. Reverse cross linking of all samples was done with reverse cross- link buffer on 65 °C at 1300 rpm for 12 h. The products were purified with QIAquick PCR purification kit (Qiagen, Hilden, Germany) following the manufacturer’s instructions. The real-time PCR was performed using SYBR Green PCR Master Mix on a sequence detector (7500 Fast Real-Time PCR System; Applied Biosystems, Foster City, CA, USA) with 2.5 μL of material per point. Primer sequences are available in Table 2. The input DNA fraction corresponded to 10% of the immunoprecipitation.

### 5.14. Statistical Analysis

All results were confirmed in three independent experiments, if not otherwise stated. The patient data are presented as median and 95% confidence interval, if not otherwise stated. All other data are presented as mean with SD. Statistical data analysis was performed using GraphPad Prism software version 8.0 (GraphPad Software, San Diego, CA, USA). Differences between groups were calculated with unpaired *t*-test. For multiple comparisons, statistical difference was calculated by one-way ANOVA. When statistically significant differences were shown, post hoc analysis were performed using the Sidak test. *p* < 0.05 was considered statistically significant.

## Figures and Tables

**Figure 1 ijms-22-07226-f001:**
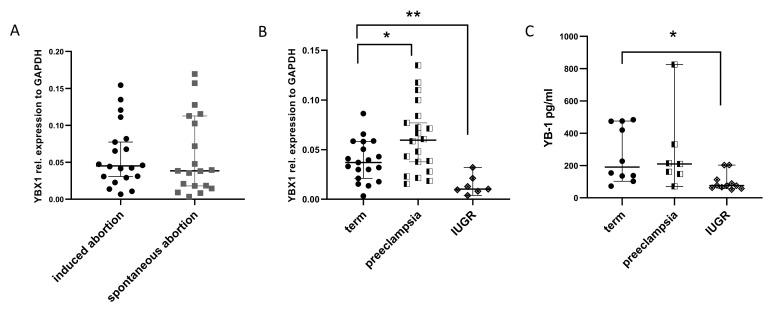
Comparison of YB-1 levels in pregnancy-associated disorders. mRNA levels of YBX1 in placenta samples from induced and spontaneous abortions (**A**), mRNA levels of YBX1 in placenta samples from term, preeclamptic, and IUGR pregnancies, at delivery (**B**), and serum YB-1 concentration comparison between term, preeclamptic, and IUGR pregnancies, at delivery (**C**). Data are presented as median with 95% confidence interval, and *p*-values were calculated with one-way ANOVA; * *p* < 0.05, ** *p* < 0.01. IUGR=intrauterine growth restriction.

**Figure 2 ijms-22-07226-f002:**
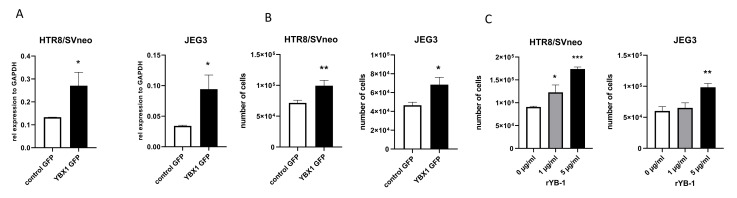
Proliferative advantage in trophoblast cell lines upon exposure to YB-1. mRNA levels of YBX1 in HTR8/SVneo and JEG3 cells 24 h after plasmid overexpression (**A**). Trophoblast cell line proliferation upon YBX1 plasmid overexpression (**B**) and after 24-h exposure to rYB-1 (**C**). The results from three independent experiments were statistically analyzed using *t*-test (**A**,**B**) or one-way ANOVA (**C**). Data are presented as mean with SD; * *p* < 0.05; ** *p* < 0.01; *** *p* < 0.001.

**Figure 3 ijms-22-07226-f003:**
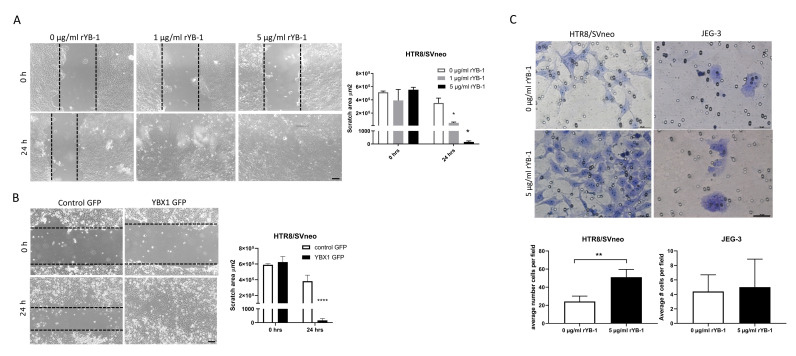
Positive effects of YB-1 on migration and invasion in HTR8/SVneo cells. Migration of HTR8/SVneo cells upon rYB-1 stimulation (**A**) and after YBX1 plasmid overexpression (**B**). Invasion rates of HTR8/SVneo and JEG3 upon 18-h rYB-1 stimulations (**C**). The results from three independent experiments were statistically analyzed using one-way ANOVA (**A**) or *t*-test (**B**,**C**). Data are presented as mean with SD; * *p* < 0.05; ** *p* < 0.01; **** *p* < 0.0001. Scale bar 50 μm.

**Figure 4 ijms-22-07226-f004:**
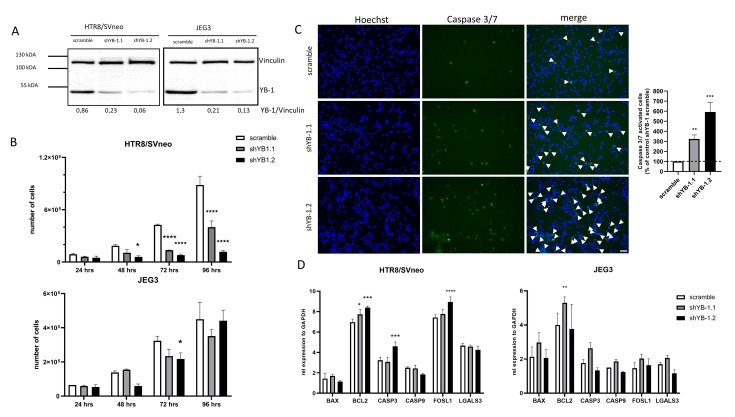
Trophoblast cell lines disadvantage upon lentiviral YB-1 downregulation. Representative Western Blot image of YB-1 protein levels 7 days upon YB-1 downregulation (**A**). Effects of YB-1 downregulation on proliferative rates (**B**), apoptosis in HTR8/SVneo cells (**C**), and apoptosis related genes expression (**D**) in trophoblast cell lines. The results from three independent experiments (except A) were statistically analyzed using one-way ANOVA. Data are presented as mean with SD; * *p* < 0.05; ** *p* < 0.01; *** *p* < 0.001; **** *p* < 0.0001. Scale bar 50 μm. BAX= BCL2-associated X, BCL2 = B-cell lymphoma 2, CASP3 = Caspase 3, CASP9 = Caspase 9, FOSL1= Fos-related antigen 1, LGALS3= Galectin-3.

**Figure 5 ijms-22-07226-f005:**
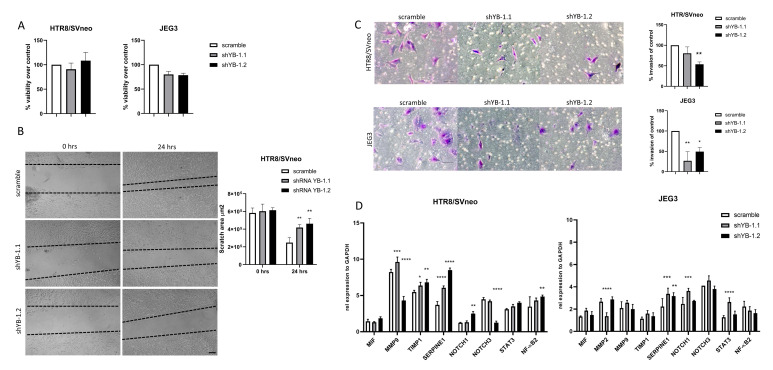
Negative effects of YB-1 on migration and invasion in HTR8/SVneo cells. YB-1 downregulation effects on trophoblast cell line viability (**A**), migration (**B**), invasion (**C**), and expression levels of genes relevant for trophoblast migration, invasion, and overall function (**D**). The results from three independent experiments were statistically analyzed using one-way ANOVA. Data are presented as mean with SD; * *p* < 0.05; ** *p* < 0.01; *** *p* < 0.001; **** *p* < 0.0001. Scale bar 50 μm. MIF = Macrophage migration inhibitory factor, MMP2 = Metalloproteinase 2, MMP9 = Metalloproteinase 9, TIMP1 = Metallopeptidase inhibitor 1, SERPINE1 = Serpin family e member 1, NOTCH1 = Notch homolog 1, NOTCH3 = Notch homolog 3, STAT3 = Signal transducer and activator of transcription 3, NF-κB = Nuclear factor kappa-light-chain-enhancer of activated B cells.

**Figure 6 ijms-22-07226-f006:**
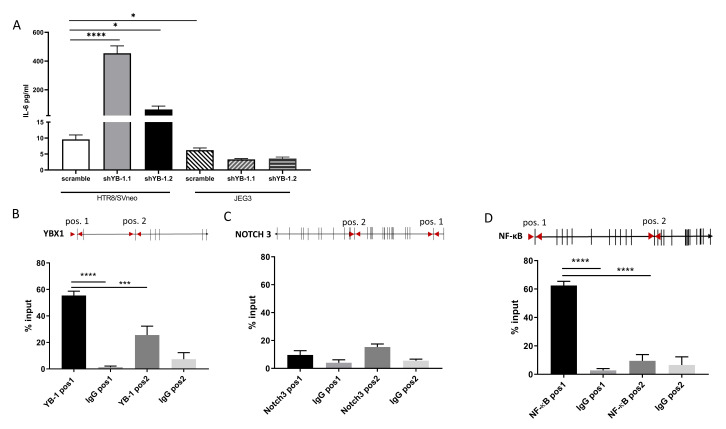
YB-1 downregulation increases IL-6 secretion and binds to NF-κB in HTR8/SVneo cells. IL-6 was detected in cell supernatants from YB-1 downregulated trophoblast cell lines (**A**). qPCR-CHIP primer positions and YB-1 binding is presented as percent of input for YB-1 (**B**), NOTCH3 (**C**), and NF-κB (**D**). The results from three independent experiments were statistically analyzed using one-way ANOVA. Data are presented as mean with SD; * *p* < 0.05; *** *p* < 0.001; **** *p* < 0.0001. NOTCH3 = Notch homolog 3, NF-κB = nuclear factor kappa-light-chain-enhancer of activated B cells.

**Table 1 ijms-22-07226-t001:** Descriptive representation of patient data. Data are presented as mean ± SD for continuous data and as % for categorical data. Spontaneous abortions were compared to induced abortions, * *p* < 0.05. IUGR and PE pregnancies were compared to control pregnancies; * *p* < 0.05; ** *p* < 0.01; **** *p* < 0.0001. PE = preeclampsia, Y = years, SC = Cesarean section.

Patient Characteristics	Induced Abortion (n = 20)	Spontaneous Abortion (n = 20)	
Maternal age (y)	26.79 ± 5,52	29.85 ± 6.84	
Gestational week	10.74 ± 1.50	9.69 ± 3.55*	
**Patient Characteristics**	**Control (n = 18)**	**IUGR (n = 12)**	**PE (n = 18)**
Maternal age (y)	28.80 ± 5.25	28.00 ± 3.07	32.29 ± 8,18
Gestational week	39.65 ± 1.58	32.25 ± 4.09 ****	28.97 ± 3,01 ****
Fetal weight (g)	3518 ± 459.3	1897 ± 869.2 **	1017 ± 388,1 ****
Fetal length (cm)	52.13 ± 2.56	46.17 ± 6.24	42.50 ± 2.12 *
Sex (% female)	33.33	63.64	42.86
Mode of delivery (% SC)	40	63.64	100

**Table 2 ijms-22-07226-t002:** qPCR-CHIP primer sequences.

No.	Primer (Gene) Name	(Sequence 5‘-3‘)
1	hYBX1 fw position 1	GGGAAGCCTTTTCTTCACGG
2	hYBX1 rv position 1	GAGTAGTCGGCCACGAAAAC
3	hYBX1 fw position 2	GAAGCTAGGGATTGGGGTCA
4	hYBX1 rv position 2	GCTACCGATCGAACTAGCGA
5	hNOTCH3 fw position 1	CACAGAGGAAGTGGGTTGCT
6	hNOTCH3 rv position 1	ATTTGCAGCCTCAGACCTCA
7	hNOTCH3 fw position 2	ATGGGGAAACACGAGAGGTTG
8	hNOTCH3 rv position 2	TTTGTCACTTGGGCCTGGGG
9	hNF-κB1 (p50) fw position 1	CCCCTCTGCCAGATCAGTATT
10	hNF-κB1 (p50) rv position 1	CGACTTGTGCCCAGTAAAGT
11	hNF-κB1 (p50) fw position 2	CTTCCTCATTCCTGCGCTAAC
12	hNF-κB1 (p50) rv position 2	GTAAGAGTTCCCCTCCGGTT

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
