# Peer review of "YB-1 Is Altered in Pregnancy-Associated Disorders and Affects Trophoblast in Vitro Properties via Alternation of Multiple Molecular Traits"

_ijms, 2021, doi:10.3390/ijms22137226_

Round 1
Reviewer 1 Report
In the manuscript entitled "YB-1 is altered in pregnancy-associated disorders and affects trophoblast in vitro properties via alternation of multiple
molecular traits", Stojanovska et al. reported the effects of YB-1 on two trophoblast cell lines HTR8/SVneo and JEG3 in terms of proliferation, migration and apoptosis. The topic is interesting and the manuscript well-written, however some cell images (Figure 3C and Figure 5C) are not very clear and must be improved to be suitable for publication.
Reviewer 2 Report
The authors have tried to explore the significance of Cold shock Y-box binding protein-1 (YB-1), a protein significant in cancer proliferation and progression, in trophoblast invasion and migration. This could shed light on the role of YB1 in pregnancy disorders such as preeclampsia.
General comments:
Considering YB1 used in the overexpression analyses is GFP-tagged and considering that cytoplasmic YB1 can promote EMT, is it possible to use the GFP signal to analyze cellular location and the difference in cytoplasmic/nuclear ratio of YB1?
For translational purposes of the study, it would be highly recommended to perform immunostaining of YB1 in patient samples and compare protein levels and localization in trophoblasts in the placenta samples.
The authors claim that YB1 increases trophoblast invasive and migratory capacity and show increased YB1 RNA levels in preeclamptic placentas. Considering preeclampsia is known to be associated with defective trophoblast invasion and migration, it further confirms the importance of showing the protein levels in the placenta samples and have a discussion on how the increase YB1 RNA may relate to decreased trophoblast invasion in preeclampsia.
As the authors point out in the discussion, HTR-8/SVneo and HEG3 cell lines are very different. HTR-8/SVneo cell line has been shown to contain a mix of trophoblastic and mesenchymal cells (https://www.sciencedirect.com/science/article/pii/S0143400416306567?via%3Dihub) while JEG3 cell line is only trophoblastic. Considering majority of the changes observed in this study are significantly pronounced in the HTR-8/SVneo cell line vs JEG3 cell line, how are the authors certain about the effects they have observed to be exclusive to trophoblastic cells? At least this should be discussed in the discussion.
Introduction:
Shallow trophoblast invasion is well documented in preeclampsia and preeclampsia associated with IUGR. However, a direct association with IUGR is not well known. Thus, please provide a relevant reference or replace “they” with “many” in line 24.
Lines 54- What is the importance of mentioning “YBX1+/- females,”? If the fetus is going to be null/null for such a gene, you would expect both parents to be heterozygous. And considering the focus is on trophoblasts that come from the embryo, this only makes the introduction confusing as to the potential effects of a heterozygote mother. If there are specific effects that are of interest, please clarify. Otherwise, please remove unnecessary details.
Results 2.1.:
- Why was gene expression analysis preferred over protein expression analysis (via western blot or immunostaining) considering in the methods section, it is mentioned that both RNA and protein extraction was performed on patient placenta samples. This is mainly significant as the protein is the “functional” component that regulates all the functional aspects from gene expression to cell invasion and migration while the RNA only shows differential YBX1 gene expression which is not necessarily correlated with protein levels or function.
- Was the RNA quality analyzed before running qPCR analysis? Is there any RIN values or any other data that can be presented to indicate that the decrease in RNA expression is not due to RNA degradation but actual gene expression?
- Based on the introduction, it is unclear why YB1 protein levels were analyzed in maternal serum? Does maternal expression of YB1 affects placentation? Or is it a matter of trophoblastic YB1 shedding into the maternal circulation?
Results 2.2.:
- The cell viability/ death data is “not shown” but the authors claim that cell “Viability” increases by YB-1. Please provide the relevant data so it can be assessed.
- Why was YB1 provided as a recombinant protein in cell culture? Is it supposed that YB1 is taken up by cells?
- If YB1 is supposed to increase cell invasion, why was YB1 added to the lower chamber in the invasion analysis (seemingly acting like a chemoattractant) instead of the upper chamber (to be taken up by the cells and promote their invasion to the lower chamber)?
- Can it be proved that YB1 was taken up by the cells?
Results 2.4.:
- Considering the differences in the effect of YB1 on the two cell lines, as shown in other panels in figure 4, panel C requires Capase3/7 activation for both cell lines.
Results 2-6:
- The CHIP-qPCR is very confusing. If CHIP-qPCR was performed to show that YB1 regulates IL-6 transcription, why are none of the primers (in table 2) for IL-6? And If YB1 is shown to attach to regulatory DNA regions (particularly promoter) of YBX1, NOTCH3 and NF-KB based on Fig 6B and table 2, then how is it relevant to IL-6? And how can the authors claim that YB1 binds NF-KB? Because the data only shows that YB1 binds NF-KB regulatory DNA regions.
- In the title of this section, the authors claim that YB1 bind NF-KB. Is the physical interaction between YB1 and NF-KB proteins confirmed in any ways? If not, please modify the title and the section accordingly.
Minor comments:
Part of Table 1 is outside the margin of the manuscript and the page.
General language editing is recommended: minor issues such as in line 172: is altered, or parenthesis missing in table 2 (Sequence 5‘-3‘).
Round 2
Reviewer 1 Report
No further comments
Author Response
We thank the reviewer for agreeing that the manuscript is set for publication.
Reviewer 2 Report
The authors investigate the role of YB1 in trophoblast migration and invasion using in vitro techniques and explore its relevance to preganncy disorders sucu as preeclampsia.
Comments:
Line 5: please specify YB1 mRNA
Line 61: mRNA expression
Results 2.1: please choose and constantly use “YB1 mRNA expression” or “YBX1 expression”. Altering between YB1 and YBX1 is confusing.
Results 2.4: The authors have replied that they do not have the Caspase 3/7 activity assay data for JEG3 cell line. But the provided RNA expression analyses in Fig 4D does not support increased apoptosis in JEG3 cell line as there is no change in expression of genes such as Caspase 3 and the only significant change is in BCL2 which can be both anti- or pro-apoptotic. How can the provided data support increased apoptosis in YB1 knock down JEG3 cell line?
Results 2.6: The authors show that YB1 can bind NF-KB enhancer and regulate its expression. Considering the NF-KB regulation of IL-6 production and activity in inflammation, the authors have evaluated IL-6 levels. However, it is unclear, and confusing, how the suggested YB1_NF-KB_IL6 pathway may be relevant considering IL-6 (Fig 6A) and NF-KB (Fige 5D) increase in YB1 downregulated HTR8/SVneo cell line.
Please check the data presentation in Table 1. It is almost impossible to follow the presented data.
Lines 207 and 208: the data only supports YBX1 expression changes in PE and IUGR and not any other pregnancy disorders. Please modify the sentence.
Methods:
Lines 311-312: Please remove protein isolation from sampling section in material and methods if the collected protein samples are inadequate to be included in the study.
Please clarify the clinical criteria used for classifying patients in preeclampsia or IUGR groups.
Please include what method has been used for comparing relative RNA expression from the qPCR data (e.g. dtCC). Additionally, please include whatever technique or method has been used to control the quality of the RNA (to show lack of degradation).
Gain and loos of function: the authors mention using gain of function or loss of function experiments while the experiments are mainly over-expression of the gene or downregulation of the mRNA (and thus protein). Please consider changing this in both abstract and discussion. It might have been relevant if the authors had used YBX1-/- cells or had expressed YBX1 in knockout cells but it is not the case.
Please correct all the references that are currently question marks as well as the many “µ” that are missing in the whole manuscript, for instance:
- Line 97: rYB-1 concentration should be 5 ug/mL.
- Line 384: 200 ul pipette?
- Line 388: um pore size
Please consider including the references and the information that YB1-GFP remains in the cytoplasm. This is significant considering the various roles and constant translocation of YB1 between the nucleus and cytoplasm as well as its high nuclear localization in cancer (particularly during cell division: https://www.ncbi.nlm.nih.gov/pmc/articles/PMC7072210/, https://www.ncbi.nlm.nih.gov/pmc/articles/PMC5556654/).
Please consider going through the language, overall.
Author Response
The authors investigate the role of YB1 in trophoblast migration and invasion using in vitro techniques and explore its relevance to preganncy disorders sucu as preeclampsia.
Comments:
Line 5: please specify YB1 mRNA
Response: Agreed and corrected.
Line 61: mRNA expression
Response: Agreed and corrected.
Results 2.1: please choose and constantly use “YB1 mRNA expression” or “YBX1 expression”. Altering between YB1 and YBX1 is confusing.
Response: We agree with the reviewer that the differently assigned gene and protein names of YB-1 might cause a confusion. However, and we hope that the reviewer will support us on this we choose to stick with the usage of the official names, so for the gene expression YBX1 and YB-1 for the protein changes.
Results 2.4: The authors have replied that they do not have the Caspase 3/7 activity assay data for JEG3 cell line. But the provided RNA expression analyses in Fig 4D does not support increased apoptosis in JEG3 cell line as there is no change in expression of genes such as Caspase 3 and the only significant change is in BCL2 which can be both anti- or pro-apoptotic. How can the provided data support increased apoptosis in YB1 knock down JEG3 cell line?
Response: Yes, that is true so we changed this accordingly in the manuscript (page 3, line 113 and page 11, lines 267-268)
Results 2.6: The authors show that YB1 can bind NF-KB enhancer and regulate its expression. Considering the NF-KB regulation of IL-6 production and activity in inflammation, the authors have evaluated IL-6 levels. However, it is unclear, and confusing, how the suggested YB1_NF-KB_IL6 pathway may be relevant considering IL-6 (Fig 6A) and NF-KB (Fige 5D) increase in YB1 downregulated HTR8/SVneo cell line.
Response: We are very sorry that this can still arise confusions. Of all the genes that we have tested in this study, only couple of ones e.g. NF-kB and Notch are known to contribute to IL-6 production. Even more the most highly induced NF-kB dependent cytokine is IL-6 (Brasier et al., 2010). We have also added this now to the manuscript page 4, line 181. In that way, we show that YB-1 downregulation leads to NF-kB upregulation that then might lead to an increased IL-6 secretion. We agree that maybe further experimentation with NF-kB downregulation might be more re-assuring. However, here we show that YB-1 and NF-kB communicate at least on a DNA level and that this in turn, might have mediated the YB-1-NF-kB-IL-6 cascade.
Please check the data presentation in Table 1. It is almost impossible to follow the presented data.
Response: We are very sorry that the representation of the data seems unclear in the final document. We have added now footnote below the table with short explanation of the data, and significance.
Lines 207 and 208: the data only supports YBX1 expression changes in PE and IUGR and not any other pregnancy disorders. Please modify the sentence.
Response: Although we understand the point of the reviewer, we would still like to keep this sentence as it is, because it is a statement that later on is discussed point-by-point in the paragraph. In the paragraph, it is clearly stated in which disorders we observe mRNA differences of YB-1 (PE and IUGR) and where not (spontaneous abortion).
Methods:
Lines 311-312: Please remove protein isolation from sampling section in material and methods if the collected protein samples are inadequate to be included in the study.
Response: Agreed and corrected.
Please clarify the clinical criteria used for classifying patients in preeclampsia or IUGR groups.
Response: We agree with the reviewer and we have added the clinical criteria to the methods section page 12-13, line 312-320.
Please include what method has been used for comparing relative RNA expression from the qPCR data (e.g. dtCC). Additionally, please include whatever technique or method has been used to control the quality of the RNA (to show lack of degradation).
Response: Agreed and added to the manuscript page 15, lines 416-418.
Gain and loos of function: the authors mention using gain of function or loss of function experiments while the experiments are mainly over-expression of the gene or downregulation of the mRNA (and thus protein). Please consider changing this in both abstract and discussion. It might have been relevant if the authors had used YBX1-/- cells or had expressed YBX1 in knockout cells but it is not the case.
Response: Agreed and corrected in the abstract and the discussion (page 1, line 7-8; and page 6, line 203-204).
Please correct all the references that are currently question marks as well as the many “µ” that are missing in the whole manuscript, for instance:
- Line 97: rYB-1 concentration should be 5 ug/mL.
- Line 384: 200 ul pipette?
- Line 388: um pore size
Response: Agreed and corrected overall in the text.
Please consider including the references and the information that YB1-GFP remains in the cytoplasm. This is significant considering the various roles and constant translocation of YB1 between the nucleus and cytoplasm as well as its high nuclear localization in cancer (particularly during cell division: https://www.ncbi.nlm.nih.gov/pmc/articles/PMC7072210/, https://www.ncbi.nlm.nih.gov/pmc/articles/PMC5556654/).
Response: We thank the reviewer for pointing out this important issue. However, as we did not focus in this study on YB-1 localization upon overexpression, we will not include these references. This will be of focus in a following study.
Please consider going through the language, overall.
Response: Agreed and we have once more checked the usage of the English language and we have found some commas and articles missing. Also, we have reformulated few sentences. We, hope now the reviewer agree with us that the language is significantly checked and improved.